# Cross-Sectional Exploration of Plasma Biomarkers of Alzheimer’s Disease in Down Syndrome: Early Data from the Longitudinal Investigation for Enhancing Down Syndrome Research (LIFE-DSR) Study

**DOI:** 10.3390/jcm10091907

**Published:** 2021-04-28

**Authors:** James A. Hendrix, David C. Airey, Angela Britton, Anna D. Burke, George T. Capone, Ronelyn Chavez, Jacqueline Chen, Brian Chicoine, Alberto C. S. Costa, Jeffrey L. Dage, Eric Doran, Anna Esbensen, Casey L. Evans, Kelley M. Faber, Tatiana M. Foroud, Sarah Hart, Kelsey Haugen, Elizabeth Head, Suzanne Hendrix, Hampus Hillerstrom, Priya S. Kishnani, Kavita Krell, Duvia Lara Ledesma, Florence Lai, Ira Lott, Cesar Ochoa-Lubinoff, Jennifer Mason, Jessie Nicodemus-Johnson, Nicholas Kyle Proctor, Margaret B. Pulsifer, Carolyn Revta, H. Diana Rosas, Tracie C. Rosser, Stephanie Santoro, Kim Schafer, Thomas Scheidemantel, Frederick Schmitt, Brian G. Skotko, Melissa R. Stasko, Amy Talboy, Amy Torres, Kristi Wilmes, Jason Woodward, Jennifer A. Zimmer, Howard H. Feldman, William Mobley

**Affiliations:** 1LuMind IDSC, 20 Mall Road, Suite 200, Burlington, MA 01803-4126, USA; abritton@lumindidsc.org (A.B.); hhillerstrom@lumindidsc.org (H.H.); 2Eli Lilly and Company, 893 Delaware St. Indianapolis, IN 46225, USA; airey_david_charles@lilly.com (D.C.A.); dage_jeffrey@lilly.com (J.L.D.); proctor_nicholas@lilly.com (N.K.P.); zimmer_jennifer_ann@lilly.com (J.A.Z.); 3St. Joseph’s Hospital and Medical Center, Department of Neurology, Barrow Neurological Institute, 350 W. Thomas Rd., Phoenix, AZ 85013, USA; Anna.Burke@DignityHealth.org; 4Down Syndrome Clinic & Research Center, Kennedy Krieger Institute, 707 N. Broadway, Baltimore, MD 21205, USA; capone@kennedykrieger.org; 5Department of Neurosciences, Alzheimer’s Disease Cooperative Study, University of California San Diego, 9500 Gilman Dr., La Jolla, CA 92093-0949, USA; chavez.ronelyn@gmail.com (R.C.); dlaraledesma@health.ucsd.edu (D.L.L.); jlmason@health.ucsd.edu (J.M.); crevta@health.ucsd.edu (C.R.); Kschafer108@gmail.com (K.S.); howardfeldman@health.ucsd.edu (H.H.F.); 6Rush University Medical Center, Division of Developmental Behavioral Pediatrics, Department of Pediatrics, 1653 W. Congress Pkwy, Chicago, IL 60612, USA; Jacqueline_Chen@rush.edu (J.C.); Cesar_Ochoa-Lubinoff@Rush.Edu (C.O.-L.); 7Adult Down Syndrome Center, Advocate Medical Group, 1610 Luther Lane, Park Ridge, IL 60068, USA; brian.chicoine@aah.org; 8Division of Neurology and Epilepsy, Department of Pediatrics, Department of Psychiatry, Case Western Reserve University School of Medicine, 10524 Euclid Ave, Cleveland, OH 44106, USA; alberto.costa@case.edu; 9Department of Pediatrics, The University of California, Irvine, 333 The City Blvd. West, Suite 800, Orange, CA 92868-4482, USA; edoran@hs.uci.edu (E.D.); itlott@hs.uci.edu (I.L.); 10Department of Pediatrics, University of Cincinnati College of Medicine, Cincinnati, OH 45229, USA; anna.esbensen@cchmc.org (A.E.); jason.woodward@cchmc.org (J.W.); 11Division of Developmental and Behavioral Pediatrics, Cincinnati Children’s Hospital Medical Center, 3333 Burnet Ave, Cincinnati, OH 45229, USA; 12Department of Psychiatry, Massachusetts General Hospital, 55 Fruit St, Boston, MA 02114, USA; CEVANS8@PARTNERS.ORG (C.L.E.); mpulsifer@mgh.harvard.edu (M.B.P.); 13National Centralized Repository for Alzheimer’s Disease and Related Dementias (NCRAD), Indiana University School of Medicine, 410 W 10th St, Indianapolis, IN 46202, USA; kelfaber@iu.edu (K.M.F.); tforoud@iu.edu (T.M.F.); wilmesk@iu.edu (K.W.); 14Duke University Medical Center, Department of Pediatrics, 2301 Erwin Road, Durham, NC 27705, USA; sarah.hart@duke.edu (S.H.); priya.kishnani@duke.edu (P.S.K.); 15Down Syndrome Program, Division of Medical Genetics and Metabolism, Department of Pediatrics, Massachusetts General Hospital, 55 Fruit St, Boston, MA 02114, USA; khaugen@mgh.harvard.edu (K.H.); KKRELL@mgh.harvard.edu (K.K.); SSANTORO3@mgh.harvard.edu (S.S.); bskotko@mgh.harvard.edu (B.G.S.); AETORRES@PARTNERS.ORG (A.T.); 16Department of Pathology and Laboratory Medicine, The University of California, Irvine, School of Medicine, 1111 Gillepsie Neuroscience Research Facility, Irvine, CA 92697, USA; heade@uci.edu; 17Pentara Corporation, 2261 East 3300 South, Suite 200, Millcreek, UT 84109, USA; shendrix@pentara.com (S.H.); jjohnson@pentaracorp.com (J.N.-J.); 18Department of Neurology, Massachusetts General Hospital, McLean Hospital, and Harvard Medical School, 55 Fruit St, Boston, MA 02114, USA; flai@partners.org (F.L.); ROSAS@HELIX.MGH.HARVARD.EDU (H.D.R.); 19Department of Human Genetics, Emory University, 615 Michael Street, Suite 301, Atlanta, GA 30322, USA; trosser@emory.edu; 20Department of Pediatrics, Harvard Medical School, 25 Shattuck Street, Boston, MA 02115, USA; 21University Hospitals Cleveland Medical Center, Department of Psychiatry, Case Western Reserve University School of Medicine, 10524 Euclid Ave, Cleveland, OH 44106, USA; Thomas.Scheidemantel@UHhospitals.org (T.S.); melissa.stasko@case.edu (M.R.S.); 22Sanders-Brown Center on Aging, University of Kentucky, 800 S. Limestone St., Lexington, KY 40536, USA; fascom@uky.edu; 23Department of Human Genetics, Emory University School of Medicine, 1365 Clifton Road, NE, Building A, Suite 2200, Atlanta, GA 30322, USA; amy.talboy@emory.edu; 24Department of Neurosciences, University of California, San Diego, 9500 Gilman Dr., La Jolla, CA 92093-0662, USA; wmobley@health.ucsd.edu

**Keywords:** Down syndrome, Alzheimer’s disease, blood biomarkers, phosphorylated tau protein, amyloid β peptide, neurofilament light chain, glial fibrillary acidic protein

## Abstract

With improved healthcare, the Down syndrome (DS) population is both growing and aging rapidly. However, with longevity comes a very high risk of Alzheimer’s disease (AD). The LIFE-DSR study (NCT04149197) is a longitudinal natural history study recruiting 270 adults with DS over the age of 25. The study is designed to characterize trajectories of change in DS-associated AD (DS-AD). The current study reports its cross-sectional analysis of the first 90 subjects enrolled. Plasma biomarkers phosphorylated tau protein (p-tau), neurofilament light chain (NfL), amyloid β peptides (Aβ_1-40_, Aβ_1-42_), and glial fibrillary acidic protein (GFAP) were undertaken with previously published methods. The clinical data from the baseline visit include demographics as well as the cognitive measures under the Severe Impairment Battery (SIB) and Down Syndrome Mental Status Examination (DS-MSE). Biomarker distributions are described with strong statistical associations observed with participant age. The biomarker data contributes to understanding DS-AD across the spectrum of disease. Collectively, the biomarker data show evidence of DS-AD progression beginning at approximately 40 years of age. Exploring these data across the full LIFE-DSR longitudinal study population will be an important resource in understanding the onset, progression, and clinical profiles of DS-AD pathophysiology.

## 1. Introduction

With improved healthcare, the Down syndrome (DS) population is experiencing a rapid increase in longevity with a life expectancy of >55 years of age compared to just 25 years of age in the 1980s. It is estimated that there are 210,000 people in the United States and about 417,000 people in Europe living with DS. Furthermore, about 40% of the DS population in the United States is over the age of 30 years [1,2]. However, with this change in longevity comes a very high risk of Alzheimer’s disease (AD). It is estimated that the lifetime risk of AD is >90% [3], and it is the leading cause of death for adults with DS [4]. DS-associated AD (DS-AD) is a slow, progressive disease associated with many of the same symptoms commonly seen in sporadic AD such as cognitive decline, behavioral issues, and functional decline impacting activities of daily living [5,6].

Early-onset AD (EOAD) is defined as those who present with clinical signs of AD before the age of 65. People with EOAD represent only about 5% or less of the 5.8 million AD cases in the United States [7] and 50 million globally [8]. A well-studied sub-set of the EOAD population are those with a known autosomal dominant AD (AD-AD) mutation in amyloid precursor protein (APP) and presenilin 1 and 2 (*PSEN1* and *PSEN2*) [9,10,11]. Individuals with DS represent the largest group of individuals with EOAD with a median age of diagnosis of AD dementia of approximately 55 years [12]. Yet people with DS remain under-studied and are often not included in the investigations seeking to generate prevalence estimates of EOAD.

DS-AD is likely driven by genetics through a gene-dose effect of APP, which is located on chromosome 21 and is overexpressed in DS [13]. Furthermore, DS-AD biomarkers follow an order of change like sporadic AD, progressing over 10–20 years before the onset of cognitive decline [14]. By their 40s, the vast majority of adults with DS develop neuropathology consistent with AD [15], while individuals with DS and partial trisomy 21 results in 2 copies of the *APP* gene serving as a notable exception. The latter support the hypothesis that an extra copy of the *APP* gene contributes to the risk of AD in people with DS [16]. There are other factors that might impact the age of onset. For example, there is evidence that people with DS who carry apolipoprotein E (*APOE*) *ε4* exhibit increased risk of earlier symptom onset and an increased amyloid load [17]. There could be other risk factors including co-morbid disorders such as obesity or untreated sleep apnea, that may also contribute to an earlier age of onset, but more research is needed to clarify their potential role.

The neuropathology of DS-AD is known to be very similar to that found in sporadic or late-onset AD (LOAD) and AD-AD. A widely supported model for sporadic and autosomal dominant AD consists of early amyloid β (Aβ) deposition, followed by hyperphosphorylated tau protein (p-tau) accumulation that is subsequently followed by neurodegeneration [18]. In this model, a long preclinical phase with the presence of AD pathology, specifically Aβ deposition, occurs more than 15 years before an individual develops overt cognitive symptoms. With the characterization of amyloid biomarkers (A), tau biomarkers (T), and neurodegeneration markers (N), it is now possible to apply both the AT(N) framework [19] and the biologically based research diagnostic criteria for Alzheimer’s disease [20]. This predominant model for LOAD and AD-AD was recently adapted to DS-AD [21,22]. Support for this model in DS-AD was seen in a recent study of a cross-sectional examination of a large cohort of adults with DS. The data from the study showed that DS-AD follows a similar trajectory of changes to that described in AD-AD. The study assessed multiple AD biomarkers (including samples from blood and cerebrospinal fluid (CSF), positron emission tomography (PET), magnetic resonance imaging (MRI), and cognitive tests) in 388 participants with DS. Data showed that biomarkers reflecting AD pathogenesis in individuals with DS behave similarly to AD biomarkers in individuals with LOAD and AD-AD [23,24].

The utility of fluid biomarkers to define the stage and progression in DS-AD is emerging. Recent data from the multi-site Alzheimer’s Biomarker Consortium–Down Syndrome (ABC-DS) cohort (*n* = 44) showed that the CSF AD biomarker profiles are very similar to LOAD and AD-AD [25]. The study showed that individuals with AD dementia had low CSF Aβ_1–42_ and Aβ_42_/Aβ_40_ ratio, indicative of amyloid deposition; increased p-tau, denoting the presence of neurofibrillary tangles [26,27,28]; and increased total tau and neurofilament light chain (NfL), indicating neurodegeneration [25]. The ABC-DS study findings are consistent with the results from the Down Alzheimer Barcelona Neuroimaging Initiative (DABNI) cohort in Europe [29]. Taken together, there is directional support for this revised AD diagnostic framework being potentially applicable and valuable in DS, with further need to characterize the sequencing and evolution of these biomarkers in DS from the asymptomatic through dementia stages [19,20]. The arrival of plasma biomarkers, which mirror CSF, and PET imaging measures provides a very attractive and more applicable less invasive fluid biomarker.

Data from the DABNI study and the LonDownS (London Down Syndrome) Consortium both point to NfL as a prognostic biomarker [29,30]. CSF Aβ_42_/Aβ_40_ ratio and plasma NfL were the first biomarkers to show changes at 28–30 years of age [23]. Use of NfL as a biomarker of LOAD has been problematic as it is a general marker of neurodegeneration and is itself correlated with age. Because people with LOAD are older than 65 and often show evidence of multiple comorbidities, the interpretation of increased NfL can be difficult. In fact, there is evidence that multiple pathologies are present together with those characteristics of AD in approximately 66% of the LOAD population. Given that aging is associated with presence of several neurodegenerative and non-degenerative diseases that also increase NfL levels, the increases in NfL cannot be attributed specifically to AD [31]. In contrast, EOAD populations, such as DS-AD, are much less likely to harbor other degenerative disorders. Thus, in EOAD, NfL can be used to more confidently define the onset of AD. In fact, emerging data indicate that plasma NfL could be useful as a prognostic biomarker in the unique DS-AD population [32]. Glial fibrillary acidic protein (GFAP) is an intermediate filament protein found in the astroglial cytoskeleton and is another emerging marker of neurodegeneration. GFAP expression is key to astrocyte function and is upregulated in neurodegenerative diseases such as AD [33]. However, more research is needed on the use of plasma GFAP as a biomarker of DS-AD progression. It appears that our study is the first report of plasma GFAP data from DS individuals in the literature. A recent study with samples from three different cohorts that included LOAD and AD-AD participants showed that plasma p-tau 217 can distinguish AD from other neurodegenerative diseases [34]. The use of plasma p-tau 217 in DS may have some specificity in being able to further define the progression and clinical diagnosis of DS-AD.

Given the strong genetic risk of AD in DS and the biomarker evidence of a temporal sequence of pathology similar to that in LOAD, a close exploration of the natural history of DS-AD should enrich our understanding of all forms of AD and enhance knowledge of potential biomarkers and outcome measures for therapeutic interventional studies. The LIFE-DSR study [35] is a longitudinal natural history study recruiting 270 adults with DS over the age of 25. The LIFE-DSR study has been designed to better understand the factors that underlie symptoms, progression, and age-related changes in the clinical presentation of DS-AD. Its design aims at characterizing trajectories of change in DS-AD at 3 visits across 32 months (baseline, month 16, and month 32) and collects sociodemographic characteristics, medical history, physical exam, neuropsychiatric evaluation (not reported here), cognitive status assessments, and a blood draw for AD biomarkers including *APOE ε4* analyses.

## 2. Experimental Section

Enrollment in the study started at the end of 2019. Paused in 2020 due to the COVID-19 pandemic, an opportunity was provided to conduct an exploratory analysis of the first 90 banked plasma samples to assess the cross-sectional distribution of baseline AD biomarkers p-tau 181, p-tau 217, NfL, Aβ_1-40_, Aβ_1-42_, and GFAP. Statistical associations between clinical measures, sociodemographic characteristics, and biomarkers were evaluated.

### 2.1. Cognitive Assessments

The Severe Impairment Battery (SIB) [36] assesses the skills of people with severe dementia utilizing 40 simple one-step commands and gestural cues, presented in a relatively naturalistic, conversational style. There are six subscales: attention, orientation, language, memory, visuospatial ability, and construction. Assessment of praxis, social interaction, and orientation to name is also included. The maximum score is 100, with higher score indicating less impairment.

Down Syndrome Mental Status Examination (DS-MSE) is a neuropsychological test battery measuring a broad range of skills including recall of personal information, orientation to season and day of the week, short-term memory, language, visuospatial construction, and praxis [37]. The maximum score is 87 using scoring method 2, with a higher score indicating less impairment [38].

### 2.2. Genetic Assays

#### Apolipoprotein E (*APOE*)

*APOE* genotypes for rs7412 and rs429358 were assayed from LIFE-DSR participant DNA by the National Centralized Repository for Alzheimer’s Disease and Related Dementias (NCRAD) to generate *APOE ε2/ε3/ε4* alleles. Genotyping was done using a customized 96-SNP chip with the Fluidigm microfluidics-based DNA fingerprinting, from DNA samples banked at NCRAD. Fingerprint data were evaluated for call rate, heterozygosity ratio, subject sex, and concordance with any other available samples; only samples passing all quality control were used to generate *APOE* genotypes.

### 2.3. Biomarker Assays

#### 2.3.1. Plasma P-tau Assays

Both p-tau181 and p-tau217 levels were measured in duplicate by electrochemiluminescence using a proprietary assay developed by Lilly Research Laboratories as previously described [25,26,27,34,39,40,41]. Briefly, samples were diluted 1:2 and 50 μL of diluted sample was used for each replicate. The assay was performed on a streptavidin small spot plate using the Meso Scale Discovery platform. P-tau181 used Biotinylated-AT270 (mIgG1) as the capture, and p-tau217 used Biotinylated-IBA493 (mIgG1) as the capture. In this study, both assays used SULFO-4G10-E2 (anti-tau monoclonal antibody developed by Lilly Research Laboratories) as the detector. Each assay was calibrated using a unique synthetic p-tau peptide.

#### 2.3.2. NfL and 4-Plex Assays

Quanterix Simoa NfL and Neurology 4-Plex E (NfL, Aβ_1-40_, Aβ_1-42_, and GFAP) assays were allowed to equilibrate to room temperature before use. Plasma samples were thawed on wet ice and briefly vortexed prior to dilution and spun at 2000× *g* for 10 min at 4C. Prediluted calibrators were also thawed on wet ice prior to use along with provided control samples. Plasma samples as well as controls were diluted 1:4 in the provided sample diluent. Further, 200 μL of all calibrators, control samples, and plasma samples were loaded onto a 450 μL v-bottom plate and loaded onto the Simoa HD-X along with kit provided SBG, beads, detector, and RGP. Each assay was run according to the manufacturer assay definitions. Since NfL results were available from both single and multiplex assays and the correlation coefficient was very high (R^2^ = 0.92), single plex Simoa NfL data were used in all analyses [32].

### 2.4. Statistical Analyses

Spearman rank-based correlations were used to test monotonic associations between biomarker and clinical variables. Associations between biomarkers and age were further investigated using linear regression with restricted cubic spline fits on raw or z-transformed scales and Huber–White robust standard errors. A secondary analysis of the association between biomarkers and age considered *APOE ε4* carrier status and sex as additive covariates. An alpha of 5% was used for statistical significance unless otherwise noted. Confidence intervals (CI) are 95% CIs. Data management, graphics, and statistical analyses were performed using Stata v16.1 and R v4.0.3 software (including tidyverse and psych packages).

## 3. Results

Among the first 90 LIFE-DSR study participants with plasma samples accessible for analysis, where demographics were available, the mean age was 38.3 years, and 46.5% were women. Baseline demographics, plasma biomarker values, and cognitive assessment scores are summarized in Table 1. Despite the relatively young age of the participants, age was strongly correlated with biomarker levels as illustrated in Figure 1. Decline in the Aβ_42_/Aβ_40_ ratio was observed starting with participants at about 45 years of age indicative of the formation of amyloid plaques.

As indicated in Figure 1, NfL values increased steadily with increasing age. In contrast, GFAP, p-tau217, and p-tau181 values were stable until the late forties at which point, they also demonstrated increases with age. Non-standardized restricted cubic spline fits for the plasma biomarkers are illustrated in Appendix A. The significant associations with age for p-tau, NfL, and GFAP were found to be robust after adjusting for *APOE ε4* carrier status and sex (Appendix A). The increase in p-tau181 and of p-tau217 by age 40 is evidence for hyperphosphorylated tau protein 10–15 years before the average age of symptom onset.

Spearman correlations between age, plasma biomarker values, and SIB and DS-MSE scores are summarized in Appendix A. Baseline SIB and DS-MSE scores were positively correlated with each other.

Inter-plasma biomarker correlations provide further insight into the progression of DS-AD. As expected, Aβ42/40 ratio was negatively correlated with NfL and p-tau181 and strong correlations were seen between p-tau217 and p-tau181. The strong correlations seen between NfL and p-tau217 or p-tau181, respectively, and GFAP and p-tau217 or p-tau181 are similar to previous studies in LOAD where the order of biomarker changes has been investigated [42]. Spearman correlations with scatter plots are illustrated in Appendix A. All statistically significant correlations remained after test multiplicity correction by Bonferroni’s method, except the weaker associations between Aβ_42_/Aβ_40_ and NfL or p-tau181.

## 4. Discussion

Plasma protein biomarkers provide an accessible tool for elucidating the molecular processes active in DS-AD. These emerging biomarkers may be useful for prognosis and diagnosis in DS-AD. In our interim analysis, we evaluated AD biomarkers of APP metabolism (Aβ_1-40_ and Aβ_1-42_), tau tangle pathology (p-tau 181 and p-tau 217), and neurodegeneration (NfL, GFAP) at a single timepoint and showed a strong association with age. These three classes of plasma biomarkers correspond with the AT(N) research framework to define AD [20]. From the modelling undertaken, the amyloid (A) biomarker measure of Aβ_1-42_/Aβ_1-40_ appears to change earliest with an initial increase in ratio followed inflection point in the mid-1940s and then progressive lowering thereafter. The tau (T) biomarkers both p-181 and p-217 tau have a progressively upwards trajectory that becomes apparent in the early to mid-1940s, while the neurodegenerative biomarkers of GFAP and NfL are also both informative. The NfL levels appear to increase progressively from the mid-1920s, while GFAP has its upward inflection point from the mid-1940s. In the initial AT(N) proposal, these neurodegenerative biomarkers were not included, however, their potential utility in DS was appreciated. The steady increase in NfL with increasing age may be particularly useful for assessing overall disease progression [32]. Overall, we searched for correlations between biomarkers and with age, *APOE ε4* status and clinical assessments of cognition and function. As highlighted above, this interim analysis of plasma biomarkers showed a strong age association. 

The use of plasma protein biomarkers, while encouraging, may have limitations. For example, the CSF Aβ_42_/Aβ_40_ ratio shows an initial decline at 28–30 years of age in DS [23]. Furthermore, evidence of amyloid deposition in people around the age of 30 is consistent with previous studies of DS brain pathology [15]. Why the Aβ_42_/Aβ_40_ measures in the CSF and plasma differ is not clear; however, one speculation is that the recently created plasma Aβ_1-40_ and Aβ_1-42_ assays lack sensitivity to detect Aβ deposition in people with DS before the age of 45.

Correlations were not observed between either the SIB or DS-MSE and the plasma biomarkers. This may be due to wide variability in cognition and function in people with DS. It may also speak to the need for more sensitive assessments designed specifically for DS-AD and validated for use in longitudinal studies. A recent longitudinal study showed an association with amyloid positron emission tomography (PET) imaging with cognitive decline as measured by the Cued Recall Test [43]. In addition, a novel composite scale has been developed with longitudinal data from multiple sites which also has the potential to detect early stages of DS-AD [44]. The LIFE-DSR study also plans to collect longitudinal cognitive and functional data and blood samples at 16 and 32 months with the hope of observing associations between plasma biomarker progression and cognitive and functional decline.

Our interim analysis has provided valuable data on the use of plasma biomarkers to help characterize DS-AD. While the number of participants was modest (*n* = 90) and the mean age is relatively young (38.3 years), the modeling of the biomarkers provides useful insights into the potential evolution of DS-AD. Remarkably, given that our analysis was cross-sectional, the biomarkers were exceptionally well behaved. They suggest that age alone predicts biomarker status and, therefore, the state of AD progression. Taken together, biomarker data are evidence that progression of DS-AD begins no later than the early forties i.e., 10–15 years before dementia is diagnosed. The next steps for the LIFE-DSR study include increasing the number of participants to 270 and gathering longitudinal data at 16 and 32 months. Longitudinal data from 270 participants should provide even greater insights into the progression of DS-AD and in so doing set the stage for clinical trials of potentially effective treatments for DS-AD and possibly for all forms of AD.

## 5. Patents

Subject matter relating to the assays, methods, reagents and/or compositions of matter set forth herein are subject to patents and/or patent applications of Eli Lilly and Company.

## Figures and Tables

**Figure 1 jcm-10-01907-f001:**
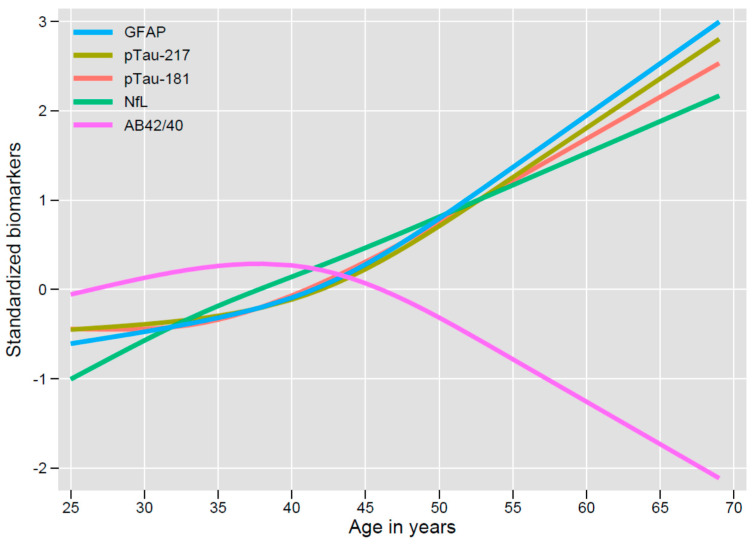
Shown are the predictions from linear regressions of each biomarker on age, using restricted cubic splines with 4 knots. Biomarkers were z-transformed to standardize the Y-axis. Robust standard errors were used. F-tests for the association of each biomarker with age was statistically significant (GFAP: F(3, 82) = 14.05, *p* < 0.0001; *p*-tau 217: F(3, 82) = 9.91, *p* < 0.0001; *p*-tau 181: F(3, 82) = 10.14, *p* < 0.0001; NfL: F(3, 82) = 34.57, *p* < 0.0001; Aβ_42_/Aβ_40_ ratio: F(3, 82) = 4.71, *p* = 0.0044), but for the Aβ_42_/Aβ_40_ ratio, this is dependent on the oldest participant; see Appendix A. (Abbreviations: Aβ_42/40_ = amyloid β_42/40_ ratio, GFAP = glial fibrillary acidic protein, NfL = neurofilament light, p-tau 181 = phosphorylated tau at threonine − 181, p-tau 217 = phosphorylated tau at threonine−217).

**Table 1 jcm-10-01907-t001:** Demographics and baseline biomarkers for first 90 subjects in Longitudinal Investigation for Enhancing Down Syndrome Research (LIFE-DSR) Study.

Characteristic	Mean (SD)	Median (Q1, Q3)	Min, Max	N (%)
p-tau 217 pg/mL	0.26 (0.38)	0.14 (0.08, 0.25)	0.01, 2.72	90 (100.00%)
p-tau 181 pg/mL	1.22 (1.10)	0.81 (0.59, 1.45)	0.37, 7.40	90 (100.00%)
NfL pg/mL	13.84 (7.19)	11.90 (9.28, 18.06)	5.13, 46.87	90 (100.00%)
GFAP pg/mL	100.37 (57.02)	84.74 (63.44, 123.65)	36.19, 445.03	90 (100.00%)
Aβ_1-40_ pg/mL	210.03 (52.36)	215.82 (188.07, 234.87)	66.32, 417.74	90 (100.00%)
Aβ_1-42_ pg/mL	8.81 (2.42)	8.90 (7.36, 10.24)	2.22, 13.82	90 (100.00%)
Aβ_42_/Aβ_40_ ratio	0.04 (0.01)	0.04 (0.04, 0.05)	0.01, 0.06	90 (100.00%)
DS-MSE Total Score 2	64.23 (11.25)	65.00 (59.00, 72.00)	30.00, 82.00	79 (87.78%)
SIB Total Score	87.92 (14.04)	93.00 (86.00, 96.50)	14.00, 100.00	84 (93.33%)
Age (years)	38.31 (9.47)	37.00 (30.00, 45.00)	25.00, 69.00	86 (95.56%)
	Female	Male		
Gender	40 (46.51%)	46 (53.49%)		
	Non-Carrier	Carrier		
*APOE ε4*	67 (74.44%)	23 (25.56%)		

Aβ_1-40_ = amyloid β peptide 1-40, Aβ_1-42_ = amyloid β peptide 1-42, *APOE ε4* = apolipoprotein E ε4 allele, DS-MSE = Down Syndrome Mental Status Examination, GFAP = glial fibrillary acidic protein, Min = minimum, Max = maximum, N = number, NfL = neurofilament light, pg/mL = picogram/milliliter, p-tau 181 = phosphorylated tau at threonine-181, p-tau 217 = phosphorylated tau at threonine−217, Q1 = quartile 1, Q3 = quartile 3, SD = standard deviation, SIB = Severe Impairment Battery.

## Data Availability

The data presented in this study are available on request from the corresponding author. The data are not publicly available due to the ongoing nature of the LIFE-DSR natural history study.

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
