# Peer review of "Cross-Sectional Exploration of Plasma Biomarkers of Alzheimer’s Disease in Down Syndrome: Early Data from the Longitudinal Investigation for Enhancing Down Syndrome Research (LIFE-DSR) Study"

_jcm, 2021, doi:10.3390/jcm10091907_

Round 1
Reviewer 1 Report
In their manuscript, Hendrix et al. report on preliminary results from the LIFE-DSR Study evaluating plasma markers for Aβ, tau and neurodegeneration. These findings are important to understanding Alzheimer’s disease progression in the Down syndrome population and should be considered for publication pending the minor revisions outlined below:
Introduction:
While mentioned briefly in the discussion, the authors should discuss the AT(N) framework of AD in the introduction and highlight previous work performed to characterize AT(N) in Down syndrome with CSF and PET measures (cite Rafii et al, Alzheimer’s Dement. 2020;12:e12062. https://doi.org/10.1002/dad2.12062), and then highlight the potential of plasma as a less invasive method for early biomarker detection.
Methods:
For the regression models, the authors chose to use a restricted cubic spline fit to model the data. The authors should elaborate on why the restricted cubic spline function specifically was chosen to fit the data. Additionally, more detail is needed on the implementation of the restricted cubic spline fit. How were the locations of the knots in the restricted cubic spline fit determined?
The restricted cubic spline works to minimize uncertainty by constraining the curve to a linear pattern at the endpoints where data is sparse (observed in Supplemental Figures 2-6 above age 50). However, in the introduction, the authors refer to the model of biomarker change outlined in Jack et al. 2010, which describes biomarker trajectories as sigmoidal in shape. These sigmoidal trajectories would likely be better modeled with the logistic growth function. How would a logistic regression compare to the results from the linear regressions with this data? If logistic regressions were not performed, this can be left as a discussion point for future work in this population with a larger sample size.
Results:
The caption for Figure 1 should provide the F-statistics for each model, similar to Supplemental Figure 7.
In section 2.5, the authors state “A secondary analysis of the association between biomarkers and age considered APOE ε4 carrier status and sex as additive covariates.” The authors should provide a summary table of the regression estimates (with 95% confidence intervals and significance) for the fits with and without APOE4 and sex adjustment.
Discussion:
The authors need to further elaborate on the AT(N) framework and how their results support the use of plasma biomarkers in Down syndrome.
More elaboration is needed on the Aβ42/40 results as no associations were found between Aβ with age and cognition, and very weak associations were observed between Aβ and p-tau. Outside of ratios, were Aβ42 and Aβ40 looked at independently with these other biomarkers? How do these values compare to those reported using SIMOA in Fortea et al 2018 (https://doi.org/10.1016/S1474-4422(18)30285-0) and Startin et al 2019 (doi: 10.1186/s13195-019-0477-0)? If changes cannot be detected prior to age 45, then plasma Aβ42/40 measures would likely not be suitable for AT(N) classification in DS since CSF and PET measures can identify Aβ change 10-15 years earlier than the speculated age for plasma detection.
More discussion is needed on the p-tau, NfL and GFAP measures in Down syndrome. How does plasma p-tau compare to CSF p-tau in Down syndrome? How do p-tau, NfL, and GFAP compare with plasma measures in sporadic AD?
DS-MSE and SIB showed no associations with age, Aβ, tau or neurodegeneration, and the authors state that a more sensitive metric of cognition is needed. The authors should elaborate on alternative cognitive outcome measures suitable for use in Down syndrome. For example, Hartley et al 2020 (https://doi.org/10.1002/dad2.12096) reports that measures of episodic memory are sensitive to detect preclinical AD changes in DS while controlling for age. Episodic memory is also highly correlated with PET Aβ measures and would likely correlate with plasma biomarkers.
Author Response
Dear Editors,
We want to thank the reviewers for their thoughtful comments. We have addressed many of the suggestions with edits to the manuscript that is now revised and submitted with this letter. We wish to respond to all the comments and suggestions below.
Reviewer 1 suggested that we add more to the introduction regarding the AT(N) research framework of AD and how this is being adapted to DS-AD. We agree and have added a bit more to the introduction including the addition of the Rafii reference as suggested.
Reviewed 1 suggested some additional clarifications for the Methods section. We thank the reviewer for the comments and offer the following explanations to our statistical approach but do not feel that changes to the manuscript are needed.
We observed nonlinear relations between age and biomarkers and elected to use a common modeling technique, restricted cubic splines. Restricted cubic splines are a flexible approach to modeling nonlinearity in regression. They fit cubic predictions between knots and linear predictions outside of the first and last knot where there is often less data and where cubic fits may be poor. Four knots were chosen at the recommended percentiles of age (Harrell, F. E., Jr. 2001. Regression Modeling Strategies: With Applications to Linear Models, Logistic Regression, and Survival Analysis. New York: Springer), resulting in knots at the percentiles 5, 35, 65, and 95 (age knots = 26, 33.5, 42, 54 yrs).
Regarding the question of logistical regression, logistic regression is a method for predicting binary outcomes. While the predicted probabilities from logistic regression are sigmoidal, the Jack et al., 2010 hypothetical biomarker longitudinal profiles could be well fit by restricted cubic splines. With more data, this may be interesting to pursue further. In the current setting with age and biomarker variables, it is appropriate to use linear regression or multiple linear regression.
In the Results section, Reviewer 1 pointed out that Figure 1 lacked the F-statistics. This omission was an error and has been corrected in the revised version.
Reviewer 1 suggested providing a summary table of the regression estimates to the Results section. We disagree, because we used linear regression with age transformed by splines, the resulting fit is not characterized by a single slope and CI. Instead, we provided supplemental figures with CIs. Adding CIs into the main figure is too busy in our opinion.
Reviewer 1 again suggested elaborating on the biomarker data and the AT(N) framework in the discussion section. We agree and have added this to the discussion including an expanded discussion of the biomarker data.
Reviewer 1 suggested more discussion is needed regarding the biomarkers and how they compare with previous literature. The assays used in our study are different from the assays used by Startin et al., 2019 and Fortea et al., 2018 and values are not directly comparable as the antibody epitopes are targeting different forms of Ab peptide (1-40 or 1-42 vs x-40 or x-42). The dynamic range of Aβ 42/40 ratio is small in plasma and thus strong correlations will be very difficult to achieve (Ref: Nakamura et al., Nature 2018; Schindler et al., Neurology 2019 and Palmqvist et al., JAMA Neurology 2019). Similarly, in our LIFE-DSR study the population includes mostly asymptomatic subjects with a lower age range that limited opportunity for detecting an association of the pathology biomarkers with age and clinical measures.
Reviewer 1 pointed out the need to include more discussion on GFAP in DS. We have revised the introduction to point out, that to our knowledge, this is the first time GFAP has been analyzed in plasma from DS individuals. Additional and larger studies are needed to investigate the role GFAP might have in DS and to perform comparisons with sporadic AD. Our results are consistent with what has been reported. Chatterjee et al., Transl Psychiatry. 2021 Jan 11;11(1):27. doi: 10.1038/s41398-020-01137-1. Verberk et al., Alzheimers Res Ther. 2020 Sep 28;12(1):118. doi: 10.1186/s13195-020-00682-7
Reviewer 1 pointed out that more should be said about the lack of associations between the plasma biomarkers and cognitive assessments. We agree and have added more to this part of the discussion and included the suggested reference from Hartley, et. al.
Again, we thank the reviewers for their constructive comments that have improved our manuscript. We hope the editors will agree that we have been responsive to the Reviewer’s comments and the revised manuscript is acceptable for publication.
Reviewer 2 Report
DS people has a high risk of Alzheimer’s disease (AD) type dementia, since all displayed plaque and neurofibrillary tangle pathology by their forties that increase with age. Despite AD-pathology, the development of dementia is not universal in this group. Therefore, exploring fluid biomarkers data, which are key to define the stage and progression of AD in DS, will help us to understand the association between AD pathogenesis and clinical profiles in this understudied group. Although the authors presented very interesting findings in this cross-sectional exploratory study showing the distribution of plasma phopho-tau, NfL, GFAP, Aβ1-40, Aβ1-42 and Aβ42/Aβ40 ratio with age, and compare to cognitive using the DS-MSE and severe impairment Battery in 90 DS people, as part of longitudinal investigation for enhancing down syndrome Research (LIFE-DSR) study, appeared unfinished/incomplete. For example, results were not stated in the abstract and figures supporting findings are missed in the paper. So, authors should add the figures from the supplemental material to the paper and findings to the abstract. Furthermore, they stated that “Biomarker distributions are described and compared to historical data in non-DS populations”, but conversely these comparisons were not mention in the experimental section or results. Authors should clarify this issue. I think these suggestions will make this paper more solid and comprehensive.
Author Response
Dear Editors,
We want to thank the reviewers for their thoughtful comments. We have addressed many of the suggestions with edits to the manuscript that is now revised and submitted with this letter. We wish to respond to all the comments and suggestions below.
We agree that the results provided in this paper are incomplete. We respectfully remind the reviewer that this is an interim analysis of baseline data. While we agree that it would be more satisfying to give complete data at the conclusion of the study, we feel that this interim analysis will still be of broad interest to the Down syndrome research community. We also feel as though the figures and tables provided in the main text represent good, overall summaries of the data. Including supplemental figures and tables will be redundant. Of course, any reader with an interest will be able to access the supplemental figures and tables as desired.
Reviewer 2 also correctly pointed out some discrepancies with the abstract. We have corrected the abstract accordingly.
Again, we thank the reviewers for their constructive comments that have improved our manuscript. We hope the editors will agree that we have been responsive to the Reviewer’s comments and the revised manuscript is acceptable for publication.
This manuscript is a resubmission of an earlier submission. The following is a list of the peer review reports and author responses from that submission.